# In Vivo and In Vitro Mechanical Loading of Mouse Achilles Tendons and Tenocytes—A Pilot Study

**DOI:** 10.3390/ijms21041313

**Published:** 2020-02-15

**Authors:** Viviane Fleischhacker, Franka Klatte-Schulz, Susann Minkwitz, Aysha Schmock, Maximilian Rummler, Anne Seliger, Bettina M. Willie, Britt Wildemann

**Affiliations:** 1Department of Trauma-, Hand- and Reconstructive Surgery, Experimental Trauma Surgery, University Hospital Jena, 07743 Jena, Germany; 2Julius Wolff Institute, Charité—Universitätsmedizin Berlin, corporate member of Freie Universität Berlin, Humboldt-Universität zu Berlin, and Berlin Institute of Health, 13353 Berlin, GermanyAysha.Schmock@charite.de (A.S.);; 3BIH Center for Regenerative Therapies, Charité—Universitätsmedizin Berlin, corporate member of Freie Universität Berlin, Humboldt-Universität zu Berlin, and Berlin Institute of Health, 13353 Berlin, Germany; 4Research Centre, Shriners Hospital for Children-Canada, Department of Pediatric Surgery, McGill University, Montreal, QC H3A 0G4, Canada

**Keywords:** tendon, mechanical loading, extracellular matrix, cell culture

## Abstract

Mechanical force is a key factor for the maintenance, adaptation, and function of tendons. Investigating the impact of mechanical loading in tenocytes and tendons might provide important information on in vivo tendon mechanobiology. Therefore, the study aimed at understanding if an in vitro loading set up of tenocytes leads to similar regulations of cell shape and gene expression, as loading of the Achilles tendon in an in vivo mouse model. In vivo: The left tibiae of mice (*n* = 12) were subject to axial cyclic compressive loading for 3 weeks, and the Achilles tendons were harvested. The right tibiae served as the internal non-loaded control. In vitro: tenocytes were isolated from mice Achilles tendons and were loaded for 4 h or 5 days (*n* = 6 per group) based on the in vivo protocol. Histology showed significant differences in the cell shape between in vivo and in vitro loading. On the molecular level, quantitative real-time PCR revealed significant differences in the gene expression of collagen type I and III and of the matrix metalloproteinases (MMP). Tendon-associated markers showed a similar expression profile. This study showed that the gene expression of tendon markers was similar, whereas significant changes in the expression of extracellular matrix (ECM) related genes were detected between in vivo and in vitro loading. This first pilot study is important for understanding to which extent in vitro stimulation set-ups of tenocytes can mimic in vivo characteristics.

## 1. Introduction

Tendons play a key role in the musculoskeletal system by transmitting forces from muscle to bone, allowing joint movement. The tissue is hypocellular and its extracellular matrix (ECM) is mainly composed of collagen type I as the predominant protein. For the maintenance of its structural integrity and function, tendons require mechanical loading. Mechanical forces include tension, hydrostatic pressure, fluid shear stress, and compression, and are transduced by cells to evoke a biochemical response. These responses affect processes such as proliferation, differentiation, tissue development, and skeletal maintenance [1]. Studies have shown that immobilization during embryonic development can lead to tendon degeneration, reduced size, or impaired formation of the tendon synovial sheath [2,3,4]. In mature tendons, mechanical loading prevents the degradation of the ECM and promotes tissue homeostasis [5]. However, depending on the strain rate, loading can have different effects on the tendon. Loading within a physiological range (4–8%) induces increased collagen production and has therapeutic effects in ex vivo tendon models [6,7,8]. Overloading (>9%) has severe effects on tendons, damaging the structural integrity, increasing cell apoptosis, and upregulating inflammatory pathways [9]. Similar effects were observed in underload (<4%) ex vivo tendon models, in which the ECM was degraded and matrix metalloproteinases (MMP) were upregulated [9]. In various in vitro cell culture studies, mechanical loading was applied to tenocytes, either biaxial or uniaxial in two-dimensional set ups [10]. Compared to ex vivo tendon models, physiological loading led to elevated collagen type I production, decorin expression, and tenogenic differentiation; however, did not activate inflammatory responses during bi-and uniaxial stretching [11,12,13,14,15]. Loading with a strain below 4% induced elevated levels of inflammatory factors, and ECM degrading enzymes, such as MMP3 [16]. High magnitude strains mimicking overuse conditions induced apoptosis, expression of various angiogenic factors, as well as inflammatory factors and MMP expression in both loading conditions [17,18,19]. For in vivo models, mechanical loading has been applied through treadmill running to induce certain conditions, such as Achilles tendinopathy [20,21]. Only a few studies exist that directly compare in vitro and in vivo mechanical loading [22,23]. 

The aim of this study was to understand if an in vitro loading set up of tenocytes leads to similar regulations of cell morphology and gene expression, as in axial compressive loading of the mouse tibia and Achilles tendon. We used comparable loading parameters in both models, and the in vivo model used in this study is a well-established model to investigate the bone formation in response of mechanical loading [24,25,26,27,28,29]. 

## 2. Results

To investigate the impact mechanical loading has on the Achilles tendon in vivo, and on mouse tenocytes in vitro, the loading protocols were chosen to be similar. The number of cycles and frequency in vitro was the same as the in vivo loading; however, the strain applied to the tendon was unknown, and therefore chosen to be 4%, as it is known that this strain is in a physiological range, and histological analysis of the loaded tendons did not show signs of degeneration. We investigated markers of tenocytes, tissue remodeling, as well as adipogenic, chondrogenic and osteogenic lineage, cytokine, and integrin expression.

### 2.1. Morphological Analysis

The Hematoxylin and Eosin staining showed the typical picture of a tendon with elongated tenocytes laying between the parallel-aligned collagen bundles. The overall structural integrity was not influenced by in vivo mechanical loading (Figure 1A). Analysis of the collagen fiber alignment demonstrated an overlapping area between the control and loaded tendon (Figure 1, red area). To investigate the organization of the fibers, the variation of the angle of the fibers was determined as the difference between the maximal and minimal angle. The variation was slightly smaller in the loaded Achilles tendon compared to the non-loaded control (Figure 1C). On the cellular level, the size of the nuclear was not significantly altered by loading in vivo or in vitro (Figure 2A). The nuclear size in the in vitro control group at day 5 was almost doubled compared to the other groups with a high variability, but the difference was not significant (*p* = 0.066). Analyzing the shape of the nuclei revealed that within the in vivo or the in vitro groups, the circularity was not influenced by mechanical stimulation. Comparing the in vitro groups to the in vivo groups, the nuclei were significantly rounder after 4 h in culture, independent of the mechanical stimulation. After 5 days of loading in vitro, no difference was seen compared to the in vivo results; however, compared to the 4-h group, a small shift to an elongated shape was observed (Figure 2B). Further, no differences regarding the amount or orientation of F-actin were found in the in vitro stimulated groups compared to the respective control (Figure 2C).

### 2.2. Gene Expression

Analysis of gene expression showed that the tendon-associated markers *Scleraxis* (*Scx*) and *Tenomodulin* (*Tnmd*) were unaffected by mechanical loading in vivo. However, in vitro, the expression of *Scx* was lower after short-term stimulation compared to the 5 days of stimulation of tenocytes, but just missed significant difference (*p* = 0.052; Figure 3A,B). Changes of *Tnmd* expression were not detected. Expression of *Runt related transcription factor 2 (Runx2)*, which is associated with osteogenic differentiation, was significantly higher after in vivo loading, compared to the in vitro stimulation of tenocytes for 5 days (Figure 3C). The chondrogenic marker *sex determining region Y (SRY)-box 9* (*Sox9*) was significantly upregulated after 4 h in vitro compared to the other in vitro and in vivo loaded groups (Figure 3D). Expression of the adipogenic marker *Lipoprotein lipase (Lpl)* was not regulated by loading (Figure 3E).

Expression levels of *Col1A1* and *Col3A1* were found to be significantly increased after in vivo loading, as well as compared to the 5-day in vitro loaded group. In vivo loading also increased *Col3A1* expression compared to the 4-h in vitro loading group, but without reaching significant difference (*p* = 0.062). In vitro, the *Col1A1* and *Col3A1* expression was unaffected by short and long time loading of tenocytes (Figure 4A,B). A similar trend was observed for *Matrix metalloproteinases 2* (*MMP2*), which was significantly upregulated in vivo compared to the unloaded control, the 5-day in vitro group, and by trend to the 4-h in vitro group (*p* = 0.052). In the in vitro groups, the expression of *MMP2* and *3* was not regulated by loading. *MMP3* expression showed a similar pattern compared to *MMP2*, but was not significantly affected (Figure 4C,D). Expression of *Tissue inhibitor of metalloproteinase 1* and *2* (*TIMP1; TIMP2*) was unchanged by loading. The increase in the *TIMP2* expression at 5 days missed significance compared to the unloaded in vitro control (*p* = 0.051) and the 4-h loading group (*p* = 0.060). *Integrin α1* and *α2* (*Itga1*, *Itga2*) expression was significantly downregulated in the in vivo loaded tendons compared to the unloaded controls. Furthermore, the expression of *Itga2* was significantly lower in the 5-day in vitro stimulation group compared to the unloaded cells. In vitro loading for 4 h did not affect integrin expression (Figure 4G,H). *Elastin (Eln), tumor necrosis factor alpha (TNF-α),* and *interleukin 1 beta* (*IL-1β*) were expressed in mouse tendons, but only in very low amounts in isolated tenocytes, which could not be evaluated. In vivo, *Eln* expression was significantly increased whereas *TNF-α* expression was significantly decreased in loaded tendons compared to unloaded controls (*p* = 0.002, *p* = 0.048, respectively. The comparison of *IL-1β* expression in loaded versus unloaded tendons did not reveal significant differences.

## 3. Discussion

The ability of cells to sense mechanical loading is a fundamental process affecting tissue homeostasis, development, and repair. Physiologic forces are important to maintain tendon tissue homeostasis and adaptation, whereas unphysiological loading can lead to injuries, such as rupture or degeneration. The response of the tendon cells to adapt by anabolic or catabolic processes depends on the applied frequency, magnitude, duration, and direction. Therefore, deciphering the mechanobiology of tenocytes is a critical step to further understand the pathophysiology in a diseased tendon and the physiologic benefits of mechanical loading during tendon adaptation and healing [30,31]. In the present study, the impact of mechanical loading on the Achilles tendon of mice was assessed. In a second step, different regimens based on the in vivo protocol were applied to mouse tenocytes in vitro using equibiaxial loading.

Histological analysis of the Achilles tendons showed no alteration of the structural integrity in loaded tendons compared to the non-loaded control. The hierarchical structure with aligned collagen fibers was observed in both groups. The findings of the present study showed no signs of a degenerated tendon as the fibers were aligned and evidence of disorganization was not found [32,33,34]. Therefore, we conclude that axial compression of the tibia used in this study does not induce a degeneration process in the mouse Achilles tendon. Further morphological analysis revealed that mechanical in vivo loading did not influence the nuclear morphology in regard to size and shape, but the 2D in vitro culture had an effect on cell nuclei. The cell nuclei in the 4-h culture had a significantly rounder shape compared to the nuclei in vivo, independent of mechanical stimulation. After 5 days in vitro, a progression towards a more elongated morphology was observed, slightly more pronounced in the loaded group. This rounding of cell nuclei of tenocytes expanded in monolayer cultures was previously shown, and seems to be reverted by a 3D culturing of the cells [35]. The organization of the F-actin in the cells loaded in vitro was unchanged compared to the non-loaded control, with no major direction. Various studies have proven that different cell types are capable of sensing mechanical loading and align the cytoskeleton in the direction of uniaxial strain [36,37,38,39]. In the present study, the applied strain was equibiaxial, and theoretically comes from all directions. If the cell experiences strain from all directions, the cytoskeleton has no predominant direction to align with.

To investigate other possible effects of mechanical loading on the tendon and tenocytes, the expression of several genes responsible for cell differentiation and the ECM was analyzed. *Scx* has a well-defined role in the development of tendons during embryogenesis. *Scx* knockout mice exhibited a disordered tendon phenotype. In addition, *Scx* promotes the proliferation of tenocytes and the production of Col1 [40,41]. Another important factor involved in various processes, such as tenocyte proliferation, collagen organization, and fibril maturation is *Tnmd* [42]. In the present study, the expression of both factors was not affected in the loaded Achilles tendons in vivo. Studies showed that moderate as well as intense treadmill running is able to increase the expression of *Scx* and *Tnmd* in mice [23,43,44]. It is important to mention that they used a different loading regimen compared to the present study. Treadmill running is a well-established method to apply physiological and non-physiological strain to the Achilles tendon in rodents. In the present study, the histological analysis showed no degenerative signs and further indicated that the expression of tendon-associated markers was not influenced by the applied loading protocols. In vitro data showed, by trend, an increase of *Scx* expression from the 4-h stimulation to the 5-day stimulation. In contrast, Huisman et al. demonstrated that *Scx* expression is already upregulated after 4-h loading in human tenocytes [45]. This difference might be explained as Huisman et al. used a strain of 10% vs. 4% in the present study; in addition, the loading time differed.

Fat infiltration and ossification are pathological signs of tendon degeneration [46]. The differentiation into non-tenocytes is known to induce degenerative processes, which can later manifest as lipid depositions and calcified tissue. To test if mechanical loading of the tibia induces molecular processes that can lead to degeneration, the differentiation markers *Lpl*, *Sox9*, and *Runx2* were examined for adipogenic, chondrogenic, and osteogenic differentiation, respectively. The expression of the differentiation markers was not significantly affected in the in vivo loaded Achilles tendons. However, the expression of *Runx2* was significantly higher in vivo compared to the 5-day in vitro loading group, where loading resulted in a decreased expression. These results support our assumption that axial compression leads to no degeneration and alteration of the cells, and are in line with a previous study, where intensive treadmill running increased *Runx2, Lpl* and *Sox9* expression in mouse Achilles tendons, while moderate running had no effect on the gene expression [23]. In the in vitro set-up, the differentiation markers were mostly unaffected from loading, except *Sox9* in the 4-h group. Similar results were observed by Zhang et al. who determined no effect on gene expression at 4% strain in vitro; however, at 8% strain expression, levels were increased in tenocytes [23]. The expression of differentiation markers can be explained by the work of Bi et al. and Klatte-Schulz et al. They showed that human and mouse tendons harbor a cell population, called tendon stem/progenitor cells that exhibit stem cell characteristics, such as multipotency, self-renewal, and clonogenicity [47,48].

This study showed that the expression of *Col1A1*, *Col3A1*, and *Elastin* was significantly upregulated in the loaded Achilles tendon, which is consistent with the findings of other studies [23,49,50]. In the aforementioned studies and the present study, different methods of mechanical loading were applied to different animal species, but still, they all showed an adaptation of the Achilles tendon to mechanical loading by increasing the production of Col 1. The in vitro data showed an opposing trend wherein the expression of collagens was not affected by loading in both groups, and was significantly less expressed than after in vivo loading. The data suggest that the in vitro environment, such as loading regimen and culture conditions (e.g., coatings and media supplements) has to be adapted to further induce and mimic in vivo collagen expression. Kim et al. showed a similar trend of *Col1* expression in rat tenocytes after loading for 12 h at a 4% strain rate [22]. Other studies, however, found an increase in collagen expression in their models [23,45]. A direct comparison of the results is difficult due to the different loading regimens used, but also indicates the variability of mechanobiology. Taken together, previous studies and the present study showed that the applied strain, frequency, and duration have a major impact and evoke different cell responses in tenocytes. 

Comparing the results of the in vitro and in vivo effects to the study by Zhang et al., similar findings were observed. The effect of moderate treadmill running and tibial loading on gene expression were comparable. This was somewhat surprising considering that exercise has systemic effects, while tibial loading has only local effects on the bone and tendon. Our data indicate that a physiological stimulus evokes similar responses, independent of the type of loading. Mechanical loading of tenocytes in vitro revealed similar expression profiles at a strain rate of 4% in both studies, although frequency and loading time differed [23]. The comparability of in vitro loading studies is often limited because the loading protocols differ in strain rate, frequency, and time.

Matrix metalloproteases and tissue inhibitors of metalloproteases play an important role in the remodeling process. The balance between MMPs and TIMPs is critical for tendon integrity. MMP2 is a gelatinase, mainly degrading small collagen fragments and gelatin. The stromelysin MMP3 primarily degrades proteoglycans and glycoproteins. The activity of MMPs is regulated by TIMPs. *MMP2* expression was significantly upregulated in the loaded Achilles tendon compared to non-loaded control. Other studies also reported changes in the expression of collagens, MMPs and TIMPs in response to physical activity [20,49,51,52,53,54,55]. The overall upregulation of *MMP*s and collagen expression indicates changes in the collagen composition, potentially leading to matrix turnover. In vitro, mechanical stimulation with 4% strain had no significant impact on the regulation of *MMP* and *TIMP* expression. In contrast, Huisman et al. showed that *MMP2 and TIMP2* expression increased in human tenocytes subjected to 10% strain at a frequency of 10 Hz [55]. To confirm the direct regulatory effects, protein and activity levels of TIMPs and MMPs have to be determined. Another important MMP that plays a major role in collagen degradation is MMP1. It is known to be involved during human tendon healing and matrix remodeling [53,56,57]. In this study, MMP1 expression could not be detected neither in the tendons of the mice nor in the tenocytes of the mice. We tested several primer pairs, as well as murine control cells and tissues, but *MMP1* expression was not detected in any of the experiments. However, it was shown that murine MMP1 is involved in various types of cancer, sepsis, and arthritis in mice [58,59,60,61]. This led us to the conclusion that MMP1 has regulatory effects in different diseases in mice, but, in contrast to humans, does not play a major role in tendon homeostasis. 

Cytokines, such as interleukins, TNF-α, and interferon gamma are known to be key players in tendon disorders. They are released by the tendon stroma or immunoregulatory cells in response to mechanical stress or tissue injury, alter the cellular phenotype, and induce changes in matrix production [62]. Our in vitro data demonstrated the absence of endogenous expression of *IL-1β* and *TNF-α* in both groups. Studies show that tenocytes are able to express pro-inflammatory cytokines by themselves in response to cytokines or in co-culture with macrophages [63,64]. In vitro mechanical loading that induces an inflammatory response comparable to over-load can be used to investigate tenocyte behavior and their mechanism to cope with inflammation. 

Integrins are cell surface receptors known to connect the ECM to the actin cytoskeleton, and transmit mechanical stimuli into the cell to evoke different cell responses. The collagen binding *Itga1* and *Itga2* were both down regulated due to in vivo loading, while only *Itga2* was down regulated after 5 days in vitro loading. A study by Popov et al. found similar results after applying 5% stretch to human tendon progenitor stem cells for one day [65]. Another study by Jones et al. suggests that various integrins are involved in the interaction of the cell with ECM during load [66]. These results confirm that further research needs to be conducted to elucidate the role of integrins during mechanotransduction. 

Taken together, the results of this study show that the applied in vivo and the in vitro loading protocols resulted only in a partially comparable reaction of the cells. The used in vitro loading protocol was designed to mimic the in vivo loading. The observed cellular reaction, however, was different between in vivo and in vitro, as well as the two in vitro protocols. On the cellular level, we observed that the nuclear size was comparable between in vivo and in vitro loading; however, a more elongated nuclei shape was seen in vivo, which was lost in tenocytes during 2D in vitro culture. On the molecular level, we observed partial similarity in the expression of tendon-associated and lineage-specific markers; however, for the ECM compartment, we determined a difference in gene expression. This can be explained by the differences in the protocols, by the different behavior of cells in a tissue or cells in a 2D culture, the normal activity of the mice between the stimulation periods, or the combination of all. Designing in vitro studies that mirror the in vivo situation is a challenging task, but this pilot study opens the discussion on how to design appropriate in vitro studies. In conclusion, either of the in vitro loading protocols could mimic some aspects of in vivo loading. For future studies, in vitro loading protocols might be extended to 21 days to have a more direct comparison to the in vivo loading protocol. Additionally, it should be considered to embed the cells in a 3D environment to better mimic the physical environment of tendons in vivo. 

## 4. Limitations

In the present study, two different kinds of mechanical loading were applied. In the in vivo stimulation, axial compression of the tibia was applied, and it is assumed that it resulted in a mechanical stimulation of the Achilles tendon, and resembled a load and relieve situation through uniaxial tension. On the other hand, equibiaxial tension was applied to the cells in vitro. In addition, it has to be considered that mechanical stimulation was applied in a 3D environment in vivo, whereas, in vitro, the cells were cultured and stimulated in a 2D environment. In the 2D in vitro environment, the cells were no longer embedded in their ECM, and this might have affected the mechanoresponsiveness of the cells. Further studies should try to mimic the in vivo situation by using a 3D set-up by embedding the isolated cells in a collagen matrix. Additionally, the comparison with other studies is often limited due to different loading parameters, cell source, and culture conditions.

A second limitation of the study was that we were unable to quantify the load experienced by the tendon. In vivo strain gauging confirmed that 2000 µε was engendered at the tibial mid-shaft of young BALB/c mice (unpublished data). During in vivo loading, the muscle is thought to be inactive as a result of the anesthesia, but the Achilles tendon is likely undergoing loading as it is compressed against the foot platen. Further measurements need to be performed to determine the force that is transmitted to the tendon during in vivo tibial loading. Additionally, we were unable to determine if the effects we observed in the tendon at the molecular level were due to direct mechanical stimulation of the tendon or if they were a result of bone-tendon crosstalk. It is likely that molecular changes occuring in the bone influence tenocytes, since it is known that bone through osteocytes functions as a secretory endocrine organ [67].

In the study, two different inbred mouse strains were used: BALB/c for the in vivo study and C57BL/6J for the in vitro study. Holguin et al. compared the outcome of two tibial axial compression regimens in BALB/c and C57BL/6J mice. They found that the loading effects on cortical bone formation were similar in both mouse strains, and concluded that the adaption of the tibia to loading was more influenced by the loading regimen than the mouse strain [27]. Although the study was focused on bone, based on these findings, we conclude that the response of the tenocytes might be similar in the BALB/c and C57BL/6J mouse strains. In accordance with the Replace, Reduce, Refine (3R) rules, tenocytes were isolated from a different mouse strain than the in vivo mouse strain, due to the availability of the animals from an independent experiment, and to avoid killing animals just for cell isolation.

## 5. Materials and Methods 

### 5.1. In Vivo Mechanical Stimulation of Achilles Tendon

Mechanical loading was applied (Testbench ElectroForce LM1, Bose, Framingham, MA, USA) to 10-week old female BALB/c mice (*n* = 12). The left tibia was subject to controlled axial compression (216 cycles/day, *f* = 4Hz, -10N load, 5 days/week, 3 weeks, ε_max_ = 2000με at tibial mid-shaft determined by strain gauging). The right tibia served as the non-loaded control. During mechanical stimulation, mice were anesthetized with 2.5% isoflurane. At day 21, mice were sacrificed through cervical dislocation, and the Achilles tendons, with adjacent bone and muscle, were dissected for histological (*n* = 6) and gene expression analysis (*n* = 6). All animal experiments were performed according to the procedures and policies approved by the legal research animal welfare representative (LAGeSo Berlin, G0027/15, approved 14 April 2015).

### 5.2. In Vitro Mechanical Stimulation of Mouse Tenocytes

Achilles tendons from female C57BL/6J mice (26 weeks) were harvested and the whole tendon was used to isolate tenocytes by enzymatic digestion with Collagenase II (3mg/mL, Biochrom AG, Berlin, Germany) for 2 h at 37 °C. Cells were grown in growth medium (Dulbecco’s Modified Eagle Medium (DMEM), with 10% fetal calf serum (FCS), and 1% Penicillin/Streptomycin (P/S), all Biochrom AG, Berlin, Germany) and passaged at a confluency of 90%. At passage 3, cells were trypsinized and seeded with a concentration of 3 × 10^4^ cells/well onto Collagen type I coated plates (Flexcell International Corporation, Burlington, USA). Prior to mechanical loading, cells were starved overnight for cell cycle synchronization in growth medium supplemented with 1% FCS. Mechanical stimulation was performed with the FX-4000^TM^ Tension system (Flexcell International Corporation, Burlington, USA) with two different loading protocols: 1) 216 cycles at 4 Hz, 4% elongation per day, 5 days; 2) 216 cycles at 4 Hz, 4% elongation, 30 min rest insertions, 5 repetitions, 4 h. Non-loaded controls were cultured in the same plates, but did not receive mechanical loading. RNA isolation and staining of the cytoskeleton was performed after the last cycle.

### 5.3. Histological Evaluation

Tendon samples were washed with 70% ethanol and fixed in a 4% paraformaldehyde solution for 24 h. Samples were decalcified for 2 weeks in ethylenediaminetetraacetic acid (EDTA) and automatically dehydrated; afterwards, tendons were paraffin embedded. For histological analysis, 4 µm thin tissue sections were cut. A Hematoxylin and Eosin (H&E) staining was performed to get an overall impression of the tendon structure. Images were taken with an AxioCam MRc5 (Zeiss, Jena, Germany) and the associated software AxioVision Rel 4.8 (Zeiss, Jena, Germany). The cell nuclei from the H&E staining were automatically analyzed in regard to their size and shape using the image analysis software ImageJ (ImageJ 1.48 v, Wayne Rasband, National Institute of Health, Bethesda, MD, USA). The Sirius Red staining was used to stain the collagen fibers and visualized under a polarization microscope. In order to visualize all birefringent collagen fibers, images were taken at six different angles (0°, 15°, 30°, 45°, 60°, 75°) and subsequently, an overlay was created with the image analysis software Fiji (Fiji 1.2 v Johannes Schindelin, Max Planck Institute of Molecular Cell Biology and Genetics, Dresden, Germany). The orientation and anisotropy of the fibers was analyzed with the FibrilTool plugin for ImageJ. The score of anisotropy was defined as the following: 0 for no order (isotropic regions) and 1 for high order (anisotropic region), such as parallel fibrils [68].

### 5.4. Phalloidin/DAPI Staining

Mechanically stimulated mouse tenocytes were fixed in 4% PFA, washed, and permeabilized three times with Phosphate Buffered Saline (PBS) + 0.025% Triton. Then, the cells were incubated with a Phalloidin probe (Invitrogen, Karlsruhe, Germany) diluted 1:200 in PBS, for 1h at room temperature (RT). The cells were washed three times with PBS + 0.025% Triton and one time with Aqua Dest (Ampuwa, Bad Homburg, Germany). The nuclei of the cells were stained with DAPI (Molecular probes, Eugene, OR, USA) diluted 1:1500 in Aqua Dest for 15 min at RT. Subsequently, the cells were washed three times with Aqua Dest. The flexible membrane was cut out and placed on top of a slide. Lastly, the membrane was mounted with Fluoromount (Southern Biotech, Birmingham, AL, USA). Cells were visualized with a fluorescence microscope (Leica DMRB, Leica, Nussloch, Germany) and excited at 358 nm or 488 nm. Morphology of the nuclei was analyzed as described above. 

### 5.5. Gene Expression Analysis

Total RNA was isolated from the dissected Achilles tendon with a precooled mortar system to homogenize the tissue. Afterwards, the powder was incubated in the lysis buffer provided by the RNA isolation kit (Qiagen, Hilden, Germany). Cells from the in vitro experiment were directly lysed in the plates through the addition of the lysis buffer. Phase separation was achieved through the addition of chloroform. After centrifugation, the aqueous phase containing the RNA was extracted and further purified with the miRNeasy Mini Kit according to the manufacturer’s protocol (Qiagen, Hilden, Germany). The purity and quantity was analyzed with the Nanodrop ND1000 system (Peqlab, Erlangen, Germany). A total amount of 100 ng RNA was transcribed into cDNA with the qScript cDNA Mix (Quanta Bioscience, Beverly, MA, USA). Quantitative real-time PCR was performed with the SYBR Green Mastermix (Quanta Bioscience) using the LightCycler 480 System (Roche, Mannheim, Germany). Primers were designed with the Primer 3 software (Table 1), except the primer pair for *Scx* (Qiagen, Hilden, Germany), and tested for their specificity and efficiency. Glyceraldehyde 3-phosphate dehydrogenase (GAPDH) served as the housekeeping gene, which has proven to be unaffected by the experimental set up. The normalized gene expression was calculated using an efficiency corrected equation.
(1)$$Mean\;normalized\;expression\;\left(MNE\right)=\frac{GAPDH\;Efficiency^{GAPDH\;Ct\;mean}}{Target\;Efficiency^{Target\;Ct\;mean}}$$

Normalized gene expression is given as a fold change to the non-loaded control.
(2)$$Fold\;gene\;expression=\frac{MNE\;\left(Loaded\right)}{MNE\;\left(Control\right)}$$

### 5.6. Statistics

Statistical analysis was performed with GraphPad version 6 (GraphPad software, La Jolla California USA). The results are presented in box plots as median with 25 and 75 percentiles. The Kruskal–Wallis test and Dunn’s multiple comparison test were used to determine statistical differences between the 3 in vivo and in vitro loading groups, as well as the 3 loading groups compared to the unloaded control. For *Eln, TNF-α*, and *IL-1β* expression, the Mann-Whitney U test was performed to investigate significant differences between loaded and unloaded tendons. A *p*-value <0.05 was considered as statistical significant. 

## 6. Conclusions

The findings of the present study provide first insights into changes of the mouse Achilles tendon after axial compression of the tibia in vivo. Histological analysis revealed only slight changes in the structural integrity; however, on a molecular level, significant differences were observed in the expression of collagens and MMPs, indicating changes in the ECM. This set up allows further investigation of mechanical loading without inducing degeneration to provide knowledge on the mechanobiology of tendons. The results of the in vitro stimulation mimicking the in vivo loading regimen and showed that partial aspects, such as nuclear size and the expression of tendon-associated markers, were similar. However, the differences of gene expression in the 2 in vitro loading protocols support the importance of strain, frequency, duration, and culturing condition. This pilot study is important for the development of an in vitro stimulation set up of tenocytes that mimics in vivo characteristics. 

## Figures and Tables

**Figure 1 ijms-21-01313-f001:**
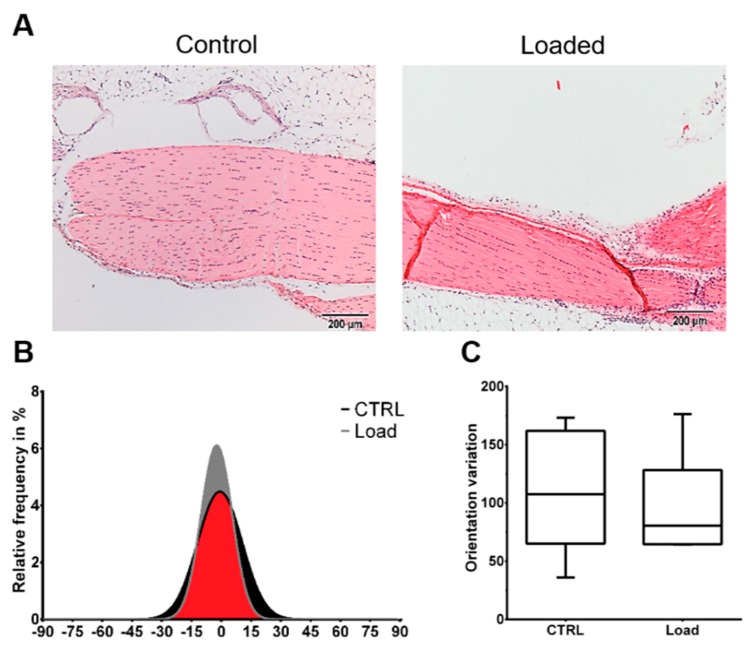
Mechanical loading did not induce histological alterations in the in vivo loaded Achilles tendon. The loaded and non-loaded Achilles tendons were dissected and further processed for histological analysis. (**A**) Representative Hematoxylin and Eosin images of the loaded and non-loaded Achilles tendon. Scale bar: 200 µm. (**B**) Collagen fiber distribution. (**C**) Variation of fiber orientation (*n* = 6).

**Figure 2 ijms-21-01313-f002:**
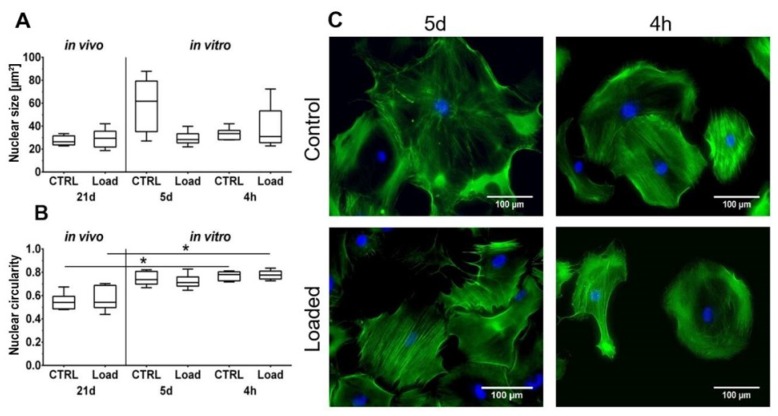
Cell morphology is partially influenced by mechanical loading. (**A**) The nuclear size is not significantly affected. (**B**) Cell nuclei in vitro were significantly rounder compared to in vivo (* *p* < 0.05, *n* = 6). (**C**) Phalloidin/ 4′,6-Diamidin-2-phenylindol (DAPI) staining of loaded murine tenocytes in vitro. Scale bar: 100 µm.

**Figure 3 ijms-21-01313-f003:**
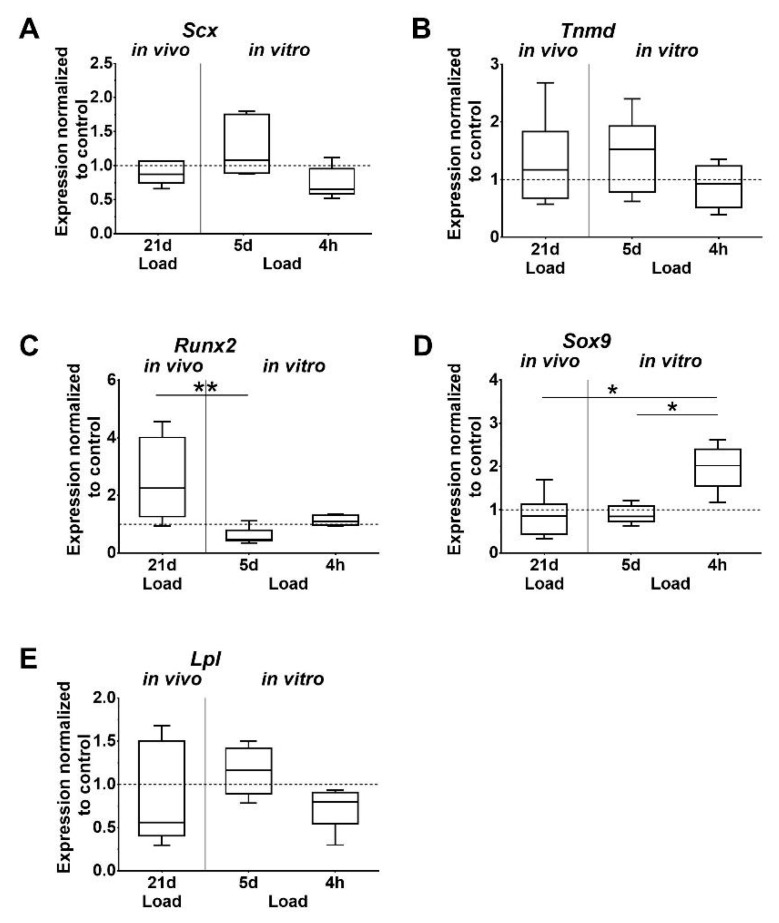
Differential expression of tissue-specific markers. (**A**,**B**) Tenogenic, (**C**–**E**) osteogenic, chondrogenic, and adipogenic markers. Gene expression is given as fold change to non-loaded controls (horizontal line). Inter-group differences are marked with an asterisk (* *p* < 0.05, ** *p* < 0.01, *n* = 6).

**Figure 4 ijms-21-01313-f004:**
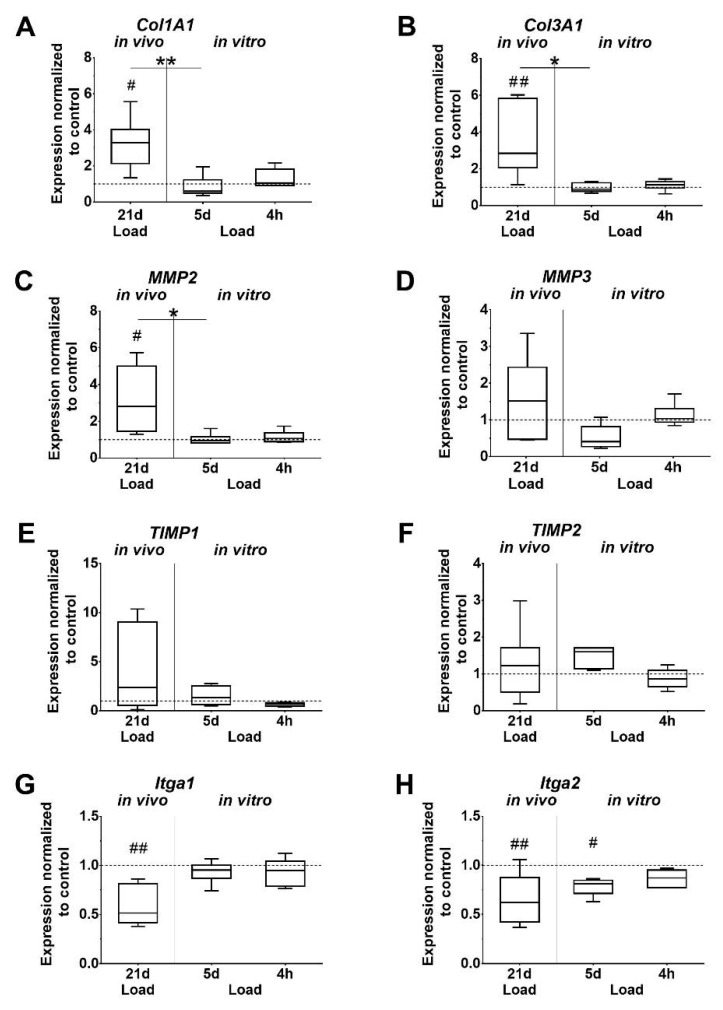
Mechanical loading influences the expression of ECM components. (**A**,**B**) Collagens, (**C**,**D**) matrix metalloproteinases (MMPs), (**E**,**F**) Tissue inhibitor of metalloproteinase (TIMPs), (**G**,**H**) Integrins. Gene expression is given as fold change to non-loaded controls (horizontal line). Significant difference to non-loaded control is marked with a pound sign and inter-group differences with an asterisk (#, *: *p* < 0.05; ##, **: *p* < 0.01, *n* = 6).

**Table 1 ijms-21-01313-t001:** List of qRT-Primers.

Gene	Accession No.	Sequence Forward Primer	Sequence Reverse Primer
*Scx*		No information provided	No information provided
*Tnmd*	NM_022322	TGTACTGGATCAATCCCACTCT	GCTCATTCTGGTCAATCCCCT
*COL1A1*	NM_007742.3	GGTCCACAAGGTTTCCAAGG	GTTCCAGGC AATCCACGAG
*COL3A1*	NM_009930.2	GCTGGAGTTGGAGGTGAAAA	GCAGCCTTGGTTAGGATCAA
*MMP2*	NM_008610.3	TGACCTTGACCAGAACACCA	TACTTTTAAGGCCCGAGCAA
*MMP3*	NM_010809.2	TGGAGATGCTCACTTTGACG	ATGGAAACGGGACAAGTCTG
*TIMP1*	NM_001044384.1	CCTTTGCATCTCTGGCATCT	CATTTCCCACAGCCTTGAAT
*TIMP2*	NM_011594.3	TACCAGATGGGCTGTGAGTG	GGGTCCTCGATGTCAAGAAA
*Runx2*	NM_009820.5	CGAAATGCCTCCGCTGTTAT	TGTCTGTGCCTTCTTGGTTCC
*Sox9*	NM_011448.4	GCAAGCAAAGGAGACCAAAA	CGCTGGTATTCAGGGAGGTA
*Lpl*	NM_008509.2	GCCAAGAGAAGCAGCAAGAT	TCCACCTCCGTGTAAATCAAG
*Itga1*	NM_001033228.3	GGCCAACCCAAAGCAAGAAC	CACATGCCAGAAATCCTCCCT
*Itga2*	NM_008396.2	GTAGTTGTGACCGATGGCGA	ACCCAAGAACTGCTATGCCG
*Eln*	NM_007925.4	CTGGTGTTGGTCTTCCAGGT	GCTTTGACTCCTGTGCCAGT
*TNF-α*	NM_013693.3	ACTGGCAGAAGAGGCACTCC	GGCTACAGGCTTGTCACTCG
*IL-1β*	NM_008361.4	TGGTGTGTGACGTTCCCATT	TCGTTGCTTGGTTCTCCTTG
*GAPDH*	NM_001289726.1	AGGTCGGTGTGAACGGATTT	TGAATTTGCCGTGAGTGGAG

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
