# Peer review of "In Vivo and In Vitro Mechanical Loading of Mouse Achilles Tendons and Tenocytes—A Pilot Study"

_ijms, 2020, doi:10.3390/ijms21041313_

Round 1

Reviewer 1 Report

In this manuscript the authors analyze and compare in vivo and in vitro protocols to understand how mechanical loading impact on tenocytes and tendons, in order to assess if an in vitro loading set up of tenocytes leads to similar regulations of cell shape and gene expression as loading of the Achilles tendon in an in vivo mouse model.

The manuscript is well written and the results clearly presented. The study is very interesting and deals with a very important topic, and the results would be very useful for researches working on tendon and tenocytes biology. However, I have many concerns especially on the experimental model designed to compare the in vivo and in vitro data and the relative results. Overall, this study has too many limitations rendering it too preliminary, and I think that further experiments should be included to complete this comparison and to provide more definitive results.

Major concerns:

1) In vitro stimulation on tenocytes in vitro could be likely influenced by the fact that cells are not immersed in extracellular matrix (ECM) as in vivo. By contrast, cells in tendon are immersed in ECM (and interact with ECM) containing collagen fibers surrounded and organized by endotenon, possibly influencing the distribution of mechanical loading and, therefore, tenocytes mechanoresponsiveness. As suggested by the authors, more reliable results could be obtained by culturing tenocytes in matrigel.

2) Results, page 2, lines 80-81: the authors report that the size of cells was not affected by loading in vivo or in vitro, but bar graphs in figure 2A shows differences between in vivo and in vitro stimulation, and between CTL and in vitro stimulated cells. Please discuss this.

3) Actin organization and arrangement in tenocytes is not evident in micrographs in figure 2C. Higher quality and/or higher magnification micrographs should be provided.

4) Results: COL gene expression is important for the analysis of COL turnover. But COL is a protein having a posttranslational regulation at different levels, therefore the gene expression analysis is not predictive of the COL secretion and content at the protein level. Moreover, COL turnover is based on COL degradation by MMPs and the MMPs/TIMPs balance is pivotal in this mechanisms. MMPs gene expression is not predictive of the COL degradation potential since MMPs are tightly regulated at the level of activation and activity. Therefore, they should be preferably studied at the protein level possibly by zymography. Moreover, the MMP starting COL-I degradation is MMP-1: to predict COL turnover, expression and activity of MMP-1 should be analyzed, while gelatinases and other MMPs play less relevant roles.

5) Discussion, page 5, line 142-143: the authors reported that “the structural integrity was not negatively influenced”. Please better and more extensively explain the meaning of “negatively”.

6) Discussion, lines 149-153: the authors refer to previous results but explanation/comments/hypotheses on this finding are not provided. Please try to discuss the result, especially focusing on the different effects exerted by the in vitro protocol.

7) Discussion, lines 171-176 and 185-192 are more a summary of previous results than a discussion: please discuss the different results obtained by in vivo and in vitro protocols providing some possible explanations.

8) Discussion, page 7, lines 233-234: I suggest to rephrase this sentence: in fact, the results do not show that the results of the vivo and in vitro protocols are not comparable, but they show that the in vitro protocol does not mimic the in vivo stimulation. I feel this is the main issue, needing further experiments to better set up the in vitro experimental conditions.

9) Materials and methods: was the mid portion of the tendon analyzed and used to extract tenocytes? This is not detailed but this is an important information, since the typical structure and composition of tendons can be studied in the mid portion.

Reviewer 2 Report

The manuscript title “In vivo and in vitro mechanical loading of mouse Achilles tendons and tenocytes – a pilot study” Here Authors  explained about Mechanical force is a key factor for the maintenance, adaption and function of tendons. Investigating the impact of mechanical loading in tenocytes and tendons might provide important information on in vivo tendon mechanobiology. Therefore, the study aimed at understanding if an in vitro loading set up of tenocytes leads to similar regulations of cell shape and gene expression as loading of the Achilles tendon in an in vivo mouse model. In vivo: The left tibiae of mice (n=12) were subjected to axial cyclic compressive loading for 3 weeks and the Achilles tendons were harvested. The right tibiae served as the internal non-loaded control. In vitro: Tenocytes were isolated from mice Achilles tendons and were loaded for 4 hours or 5 days (n=6 per group) based on the in vivo protocol. Histology showed significant differences in the cell shape between in vivo and in vitro loading. On the molecular level, quantitative real-time PCR revealed significant differences in the gene expression of collagen type I and III and moreover of MMP expression. Tendon-associated markers showed a similar expression profile. This study showed that gene expression of tendon markers was similar, whereas significant changes in the expression of ECM related genes extracellular matrix compartment were detected between in vivo and in vitro loading. These preliminary experiments confirm that  important for the understanding to which extend in vitro stimulation set-ups of tenocytes can mimic in vivo characteristics. The article is well written and the author did enough experiments to understand if an in vitro loading set up of tenocytes leads to similar regulations of cell morphology and gene expression as in axial compressive loading of the mouse tibial and Achilles tendon.It was convincing. I would prefer this article is suitable for publication but after one minor experiment.

1.Could the authors check the gene expression of TNF-alpha and IL-I  in gene expression analysis.

Round 2

Reviewer 1 Report

The authors addressed most issues and the manuscript was improved.